# COMPLEX QUERY ANSWERING
# WITH NEURAL LINK PREDICTORS

**Erik Arakelyan**[1†]**, Daniel Daza**[2,3,4†]**, Pasquale Minervini**[1†]**, & Michael Cochez**[2,4]
[1]UCL Centre for Artificial Intelligence, University College London, United Kingdom
[2]Vrije Universiteit Amsterdam, The Netherlands
[3]University of Amsterdam, The Netherlands
[4]Discovery Lab, Elsevier, The Netherlands
{erik.arakelyan.18,p.minervini}@ucl.ac.uk
{d.dazacruz,m.cochez}@vu.nl

## ABSTRACT

Neural link predictors are immensely useful for identifying missing edges in large scale Knowledge Graphs. However, it is still not clear how to use these models for answering more complex queries that arise in a number of domains, such as queries using logical conjunctions ($\wedge$), disjunctions ($\vee$) and existential quantifiers ($\exists$), while accounting for missing edges. In this work, we propose a framework for efficiently answering complex queries on incomplete Knowledge Graphs. We translate each query into an end-to-end differentiable objective, where the truth value of each atom is computed by a pre-trained neural link predictor. We then analyse two solutions to the optimisation problem, including gradient-based and combinatorial search. In our experiments, the proposed approach produces more accurate results than state-of-the-art methods — black-box neural models trained on millions of generated queries — without the need of training on a large and diverse set of complex queries. Using orders of magnitude less training data, we obtain relative improvements ranging from 8% up to 40% in Hits@3 across different knowledge graphs containing factual information. Finally, we demonstrate that it is possible to *explain* the outcome of our model in terms of the intermediate solutions identified for each of the complex query atoms. All our source code and datasets are available online [1].

## 1 INTRODUCTION

Knowledge Graphs (KGs) are graph-structured knowledge bases, where knowledge about the world is stored in the form of relationship between entities. KGs are an extremely flexible and versatile knowledge representation formalism – examples include general purpose knowledge bases such as DBpedia (Auer et al., 2007) and YAGO (Suchanek et al., 2007), domain-specific ones such as Bio2RDF (Dumontier et al., 2014) and Hetionet (Himmelstein et al., 2017) for life sciences and WordNet (Miller, 1992) for linguistics, and application-driven graphs such as the Google Knowledge Graph, Microsoft's Bing Knowledge Graph, and Facebook's Social Graph (Noy et al., 2019).

Neural link predictors (Nickel et al., 2016) tackle the problem of identifying missing edges in large KGs. However, in many complex domains, an open challenge is developing techniques for answering complex queries involving multiple and potentially unobserved edges, entities, and variables, rather than just single edges.

We focus on First-Order Logical Queries that use conjunctions ($\wedge$), disjunctions ($\vee$), and existential quantifiers ($\exists$). A multitude of queries can be expressed by using such operators – for instance, the query *"Which drugs $D$ interact with proteins associated with diseases $t_1$ or $t_2$?"* can be rewritten as $?D : \exists P.\text{interacts}(D, P) \wedge [\text{assoc}(P, t_1) \vee \text{assoc}(P, t_2)]$, which can be answered via sub-graph matching.

---

[1]At https://github.com/uclnlp/cqd
[†]Equal contribution, alphabetical order.

*"Which drugs interact with proteins associated with diseases $t_1$ or $t_2$?"*

$?D : \exists P \,.\, \text{interacts}(D, P) \wedge \big[\text{assoc}(P, t_1) \vee \text{assoc}(P, t_2)\big]$

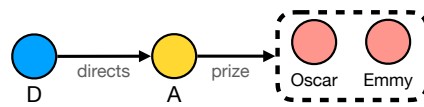

*"Which directors directed actors that won either an Oscar or an Emmy?"*

$?D : \exists A \,.\, \text{directs}(D, A) \wedge \big[\text{prize}(A, \text{Oscar}) \vee \text{prize}(A, \text{Emmy})\big]$

Figure 1: Examples of First-Order Logical Queries using existential quantification ($\exists$), conjunction ($\wedge$), and disjunction ($\vee$) operators — their dependency graphs are $D \leftarrow P \leftarrow \{t_1, t_2\}$, and $D \leftarrow A \leftarrow \{\text{Oscar}, \text{Emmty}\}$, respectively.

However, plain sub-graph matching cannot capture semantic similarities between entities and relations, and cannot deal with missing facts in the KG. One possible solution consists in computing all missing entries via KG completion methods (Getoor & Taskar, 2007; De Raedt, 2008; Nickel et al., 2016), but that would materialise a significantly denser KG and would have intractable space and time complexity requirements (Krompaß et al., 2014).

In this work, we propose a framework for answering First-Order Logic Queries, where the query is compiled in an end-to-end differentiable function, modelling the interactions between its atoms. The truth value of each atom is computed by a neural link predictor (Nickel et al., 2016) – a differentiable model that, given an atomic query, returns the likelihood that the fact it represents holds true. We then propose two approaches for identifying the most likely values for the variable nodes in a query – either by continuous or by combinatorial optimisation.

Recent work on embedding logical queries on KGs (Hamilton et al., 2018; Daza & Cochez, 2020; Ren et al., 2020) has suggested that in order to go beyond link prediction, more elaborate architectures, and a large and diverse dataset with millions of queries is required. In this work, we show that this is not the case, and demonstrate that it is possible to use an efficient neural link predictor trained for 1-hop query answering, to generalise to up to 8 complex query structures. By doing so, we produce more accurate results than state-of-the-art models, while using orders of magnitude less training data.

Summarising, in comparison with other approaches in the literature such as Query2Box (Ren et al., 2020), we find that the proposed framework i) achieves significantly better or equivalent predictive accuracy on a wide range of complex queries, ii) is capable of out-of-distribution generalisation, since it is trained on simple queries only and evaluated on complex queries, and iii) is more explainable, since the intermediate results for its sub-queries and variable assignments can be used to explain any given answer.

## 2 EXISTENTIAL POSITIVE FIRST-ORDER LOGICAL QUERIES

A Knowledge Graph $\mathcal{G} \subseteq \mathcal{E} \times \mathcal{R} \times \mathcal{E}$ can be defined as a set of subject-predicate-object $\langle s, p, o \rangle$ triples, where each triple encodes a relationship of type $p \in \mathcal{R}$ between the subject $s \in \mathcal{E}$ and the object $o \in \mathcal{E}$ of the triple, where $\mathcal{E}$ and $\mathcal{R}$ denote the set of all entities and relation types, respectively. One can think of a Knowledge Graph as a labelled multi-graph, where entities $\mathcal{E}$ represent nodes, and edges are labelled with relation types $\mathcal{R}$. Without loss of generality, a Knowledge Graph can be represented as a First-Order Logic Knowledge Base, where each triple $\langle s, p, o \rangle$ denotes an atomic formula $p(s, o)$, with $p \in \mathcal{R}$ a binary predicate and $s, o \in \mathcal{E}$ its arguments.

Conjunctive queries are a sub-class of First-Order Logical queries that use existential quantification ($\exists$) and conjunction ($\wedge$) operations. We consider conjunctive queries $\mathcal{Q}$ in the following form:

$$
\begin{aligned}
\mathcal{Q}[A] \triangleq ?A : \quad & \exists V_1, \dots, V_m . e_1 \wedge \dots \wedge e_n \\
\text{where} \quad & e_i = p(c, V), \text{ with } V \in \{A, V_1, \dots, V_m\}, c \in \mathcal{E}, p \in \mathcal{R} \\
\text{or} \quad & e_i = p(V, V'), \text{ with } V, V' \in \{A, V_1, \dots, V_m\}, V \neq V', p \in \mathcal{R}.
\end{aligned}
\tag{1}
$$

In Eq. (1), the variable $A$ is the *target* of the query, $V_1, \dots, V_m$ denote the *bound variable nodes*, while $c \in \mathcal{E}$ represent the *input anchor nodes*. Each $e_i$ denotes a logical atom, with either one ($p(c, V)$) or two variables ($p(V, V')$), and $e_1 \wedge \dots \wedge e_n$ denotes a conjunction between $n$ atoms.

The goal of answering the logical query $\mathcal{Q}$ consists in finding a set of entities $[\![\mathcal{Q}]\!] \subseteq \mathcal{E}$ such that $a \in [\![\mathcal{Q}]\!]$ iff $\mathcal{Q}[a]$ holds true, where $[\![\mathcal{Q}]\!]$ is the *answer set* of the query $\mathcal{Q}$.

As illustrated in Fig. 1, the *dependency graph* of a conjunctive query $\mathcal{Q}$ is a graph representation of $\mathcal{Q}$ where nodes correspond to variable or non-variable atom arguments in $\mathcal{Q}$ and edges correspond to atom predicates. We follow Hamilton et al. (2018) and focus on *valid* conjunctive queries – i.e. the dependency graph needs to be a directed acyclic graph, where anchor entities correspond to source nodes, and the query target $A$ is the unique sink node.

**Example 2.1** (Conjunctive Query). *Consider the query "Which drugs interact with proteins associated with the disease $t$?". This query can be formalised as a conjunctive query $\mathcal{Q}$ such as $?D : \exists P.interacts(D, P) \wedge assoc(P, t)$, where $t$ is an* input anchor node, *the variable $D$ is the* target *of the query, $P$ is a* bound variable node, *and the dependency graph is $D \leftarrow P \leftarrow t$. The* answer set $[\![\mathcal{Q}]\!]$ *of $\mathcal{Q}$ corresponds to the set of all drugs in $\mathcal{E}$ interacting with proteins associated with $t$.* ∎

**Handling Disjunctions**   So far we focused on conjunctive queries defined using the existential quantification ($\exists$) and conjunction ($\wedge$) logical operators. Our aim is answering a wider class of logical queries, namely Existential Positive First-Order (EPFO) queries (Dalvi & Suciu, 2004) that in addition to existential quantification and conjunction, also involve disjunction ($\vee$). We follow Ren et al. (2020) and, without loss of generality, we transform a given EPFO query into Disjunctive Normal Form (DNF, Davey & Priestley, 2002), i.e. a disjunction of conjunctive queries.

**Example 2.2** (Disjunctive Normal Form). *Consider the following variant of query in Example 2.1: "Which drugs interact with proteins associated with the diseases $t_1$ or $t_2$?". This query can be formalised as a EPFO query $\mathcal{Q}$ such as $?D : \exists P.interacts(D, P) \wedge [assoc(P, t_1) \vee assoc(P, t_2)]$. We can transform $\mathcal{Q}$ in the following, equivalent DNF query: $?D : \exists P. [interacts(D, P) \wedge assoc(P, t_1)] \vee [interacts(D, P) \wedge assoc(P, t_2)]$.* ∎

In our framework, given a DNF query $\mathcal{Q}$, for each of its conjunctive sub-queries we produce a score for all the entities representing the likelihood that they answer that sub-query. Finally, such scores are aggregated using a t-conorm — a continuous relaxation of the logical disjunction.

## 3   COMPLEX QUERY ANSWERING VIA OPTIMISATION

We propose a framework for answering EPFO logical queries in the presence of missing edges. Given a query $\mathcal{Q}$, we define the score of a target node $a \in \mathcal{E}$ as a candidate answer for a query as a function of the score of all atomic queries in $\mathcal{Q}$, given a variable-to-entity substitution for all variables in $\mathcal{Q}$.

Each variable is mapped to an *embedding vector*, that can either correspond to an entity $c \in \mathcal{E}$ or to a virtual entity. The score of each of the query atoms is determined individually using a neural link predictor (Nickel et al., 2016). Then, the score of the query with respect to a given candidate answer $\mathcal{Q}[a]$ is computed by aggregating all atom scores using t-norms and t-conorms – continuous relaxations of the logical conjunction and disjunction operators.

**Neural Link Prediction**   A neural link predictor is a differentiable model where atom arguments are first mapped into a $k$-dimensional embedding space, and then used for producing a score for the atom. More formally, given a query atom $p(s, o)$, where $p \in \mathcal{R}$ and $s, o \in \mathcal{E}$, the score for $p(s, o)$ is computed as $\phi_p(\mathbf{e}_s, \mathbf{e}_o)$, where $\mathbf{e}_s, \mathbf{e}_o \in \mathbb{R}^k$ are the embedding vectors of $s$ and $o$, and $\phi_p : \mathbb{R}^k \times \mathbb{R}^k \mapsto [0, 1]$ is a *scoring function* computing the likelihood that entities $s$ and $o$ are related by the relationship $p$.

In our experiments, as neural link predictor, we use ComplEx (Trouillon et al., 2016) regularised using a variational approximation of the tensor nuclear $p$-norm proposed by Lacroix et al. (2018).

**T-Norms**   A *t-norm* $\top : [0, 1] \times [0, 1] \mapsto [0, 1]$ is a generalisation of conjunction in logic (Klement et al., 2000; 2004). Some examples include the *Gödel t-norm* $\top_{\min}(x, y) = \min\{x, y\}$, the *product t-norm* $\top_{\text{prod}}(x, y) = x \cdot y$, and the *Łukasiewicz t-norm* $\top_{\text{Luk}}(x, y) = \max\{0, x + y - 1\}$. Analogously, *t-conorms* are dual to t-norms for disjunctions – given a t-norm $\top$, the complementary t-conorm is defined by $\bot(x, y) = 1 - \top(1 - x, 1 - y)$.

**Continuous Reformulation of Complex Queries** Let $\mathcal{Q}$ denote the following DNF query:

$$\mathcal{Q}[A] \triangleq ?A : \quad \exists V_1, \ldots, V_m. \left(e_1^1 \wedge \ldots \wedge e_{n_1}^1\right) \vee .. \vee \left(e_1^d \wedge \ldots \wedge e_{n_d}^d\right)$$

$$\text{where} \quad e_i^j = p(c, V), \text{ with } V \in \{A, V_1, \ldots, V_m\}, c \in \mathcal{E}, p \in \mathcal{R} \tag{2}$$

$$\text{or} \quad e_i^j = p(V, V'), \text{ with } V, V' \in \{A, V_1, \ldots, V_m\}, V \neq V', p \in \mathcal{R}.$$

We want to know the variable assignments that render $\mathcal{Q}$ true. To achieve this. we can cast this as an optimisation problem, where the aim is finding a mapping from variables to entities that *maximises* the score of $\mathcal{Q}$:

$$\underset{A, V_1, \ldots, V_m \in \mathcal{E}}{\arg\max} \left(e_1^1 \top \ldots \top e_{n_1}^1\right) \perp .. \perp \left(e_1^d \top \ldots \top e_{n_d}^d\right)$$

$$\text{where} \quad e_i^j = \phi_p(\mathbf{e}_c, \mathbf{e}_V), \text{ with } V \in \{A, V_1, \ldots, V_m\}, c \in \mathcal{E}, p \in \mathcal{R} \tag{3}$$

$$\text{or} \quad e_i^j = \phi_p(\mathbf{e}_V, \mathbf{e}_{V'}), \text{ with } V, V' \in \{A, V_1, \ldots, V_m\}, V \neq V', p \in \mathcal{R},$$

where $\top$ and $\perp$ denote a t-norm and a t-conorm – a continuous generalisation of the logical conjunction and disjunction, respectively – and $\phi_p(\mathbf{e}_s, \mathbf{e}_o) \in [0, 1]$ denotes the neural link prediction score for the atom $p(s, o)$. We write t-norms and t-conorms as infix operators since they are both associative.

Note that, in Eq. (3), the bound variable nodes $V_1, \ldots, V_m$ are only used through their embedding vector: to compute $\phi_p(\mathbf{e}_c, \mathbf{e}_V)$ we only use the embedding representation $\mathbf{e}_V \in \mathbb{R}^k$ of $V$, and do not need to know which entity the variable $V$ corresponds to. This means that we have two possible strategies for finding the optimal variable embeddings $\mathbf{e}_V \in \mathbb{R}^k$ with $V \in \{A, V_1, \ldots, V_m\}$ for maximising the objective in Eq. (3), namely *continuous optimisation*, where we optimise $\mathbf{e}_V$ using gradient-based optimisation, and *combinatorial optimisation*, where we search for the optimal variable-to-entity assignment.

### 3.1 COMPLEX QUERY ANSWERING VIA CONTINUOUS OPTIMISATION

One way we can solve the optimisation problem in Eq. (3) is by finding the variable embeddings that maximise the score of a complex query. This can be formalised as the following continuous optimisation problem:

$$\underset{\mathbf{e}_A, \mathbf{e}_{V_1}, \ldots, \mathbf{e}_{V_m} \in \mathbb{R}^k}{\arg\max} \left(e_1^1 \top \ldots \top e_{n_1}^1\right) \perp .. \perp \left(e_1^d \top \ldots \top e_{n_d}^d\right)$$

$$\text{where} \quad e_i^j = \phi_p(\mathbf{e}_c, \mathbf{e}_V), \text{ with } V \in \{A, V_1, \ldots, V_m\}, c \in \mathcal{E}, p \in \mathcal{R} \tag{4}$$

$$\text{or} \quad e_i^j = \phi_p(\mathbf{e}_V, \mathbf{e}_{V'}), \text{ with } V, V' \in \{A, V_1, \ldots, V_m\}, V \neq V', p \in \mathcal{R}.$$

In Eq. (4) we directly optimise the embedding representations $\mathbf{e}_A, \mathbf{e}_{V_1}, \ldots, \mathbf{e}_{V_m} \in \mathbb{R}^k$ of variables $A, V_1, \ldots, V_m$, rather than exploring the combinatorial space of variable-to-entity mappings. In this way, we can tackle the maximisation problem in Eq. (4) using gradient-based optimisation methods, such as Adam (Kingma & Ba, 2015). Then, after we identified the optimal representation for variables $A, V_1, \ldots, V_m$, we replace the query target embedding $\mathbf{e}_A$ with the embedding representations $\mathbf{e}_c \in \mathbb{R}^k$ of all entities $c \in \mathcal{E}$, and use the resulting complex query score to compute the likelihood that such entities answer the query.

### 3.2 COMPLEX QUERY ANSWERING VIA COMBINATORIAL OPTIMISATION

Another way we tackle the optimisation problem in Eq. (3) is by greedily searching for a set of variable substitutions $S = \{A \leftarrow a, V_1 \leftarrow v_1, \ldots, V_m \leftarrow v_m\}$, with $a, v_1, \ldots, v_m \in \mathcal{E}$, that maximises the complex query score, in a procedure akin to *beam search*. We do so by traversing the dependency graph of a query $\mathcal{Q}$ and, whenever we find an atom in the form $p(c, V)$, where $p \in \mathcal{R}$, $c$ is either an entity or a variable for which we already have a substitution, and $V$ is a variable for which we do not have a substitution yet, we replace $V$ with all entities in $\mathcal{E}$ and retain the top-$k$ entities $t \in \mathcal{E}$ that maximise $\phi_p(\mathbf{e}_c, \mathbf{e}_t)$ – i.e. the most likely entities to appear as a substitution of $V$ according to the neural link predictor.

Our procedure is akin to beam search: as we traverse the dependency graph of a query, we keep a beam with the most promising variable-to-entity substitutions identified so far.

**Example 3.1** (Combinatorial Optimisation). *Consider the query "Which drugs $D$ interact with proteins associated with disease $t$?" can be rewritten as: $?D : \exists P.interacts(D, P) \land assoc(P, t)$. In order to answer this query via combinatorial optimisation, we first find the top-$k$ proteins $p$ that are most likely to substitute the variable $P$ in $assoc(P, t)$. Then, we search for the top-$k$ drugs $d$ that are most likely to substitute $D$ in $interacts(D, P)$, ending up with at most $k^2$ candidate drugs. Finally, we rank the candidate drugs $d$ by using the query score produced by the t-norm.* ∎

Note that scoring all possible entities can be done efficiently and in a single step on a GPU by replacing $V$ with the entity embedding matrix. In our experiments we did not notice any computational bottlenecks due to the branching factors of longer queries. However, that could be handled by using alternate graph exploration strategies.

## 4 RELATED WORK

This work is closely related to approaches for learning to traverse Knowledge Graphs (Guu et al., 2015; Das et al., 2017; 2018), and more recent works on answering conjunctive queries via black-box neural models trained on generated queries (Hamilton et al., 2018; Daza & Cochez, 2020; Kotnis et al., 2020). The main difference is that we propose a tractable framework for handling a substantially larger subset of First-Order Logic queries.

More recently, Ren et al. (2020) proposed Query2Box, a neural model for Existential Positive First-Order logical queries, where queries are represented via box embeddings (Li et al., 2019). Such approaches for query answering require a dataset with *millions* of generated queries to generalise well – for instance, on the FB15k-237 dataset, approx. $15 \times 10^4$ training queries for each query type are used, resulting in approx. $1.2 \times 10^6$ training queries. Our framework, on the other hand, only uses a simple, state-of-the-art neural link predictor (Lacroix et al., 2018) trained on a set of 1-hop queries that is orders of magnitude smaller.

There is a large body of work on neural link predictors, that learn embeddings of entities and relations in KGs via a simple link prediction training objective (Bordes et al., 2013; Yang et al., 2015; Trouillon et al., 2016; Lacroix et al., 2018). Due to their design, they are often evaluated for answering 1-hop queries only, as their application to more complex queries does not derive directly from their formulation.

Previous work has considered using such embeddings for complex query answering, by partitioning the query graph and using an ad-hoc aggregation function to score candidate answers (Wang et al., 2018), or by using a probabilistic mixture model similar to DistMult (Friedman & den Broeck, 2020). In contrast, our proposed method answers a query by using a single pass where aggregation steps are implemented with t-norms and t-conorms, which are continuous relaxations of conjunctions and disjunctions. Such t-norms have been proposed as differentiable formulations of logical operators suitable for gradient-based learning (Serafini & d'Avila Garcez, 2016; Guo et al., 2016; Minervini et al., 2017; van Krieken et al., 2020).

Further alternatives for using embeddings from neural link predictors, such as combinatorial optimisation, have been ruled out as unfeasible (Hamilton et al., 2018; Daza & Cochez, 2020). We show that this approach can scale well by reducing the set of possible intermediate answers, while outperforming the state-of-the-art in query answering.

The framework proposed in this paper is related to neural theorem provers (Rocktäschel & Riedel, 2017; Weber et al., 2019; Minervini et al., 2020a;b), a differentiable relaxation of the backward-chaining reasoning algorithm where comparison between symbols is replaced by a differentiable similarity function between their embedding vectors. During the reasoning process, neural theorem provers check which rules can be used for proving a given atomic query. Then it is checked whether the premise of such rules is satisfied, where the premise is a conjunctive query. The procedure they use for answering conjunctions is akin to the combinatorial optimisation procedure we propose in Section 3.2. The main source of difference is how atomic queries are answered – we use the ComplEx neural link predictor (Trouillon et al., 2016), while neural theorem provers use the maximum similarity value between a given atomic query and all facts in the Knowledge Graph, which has linear complexity in the number of triples in the graph.

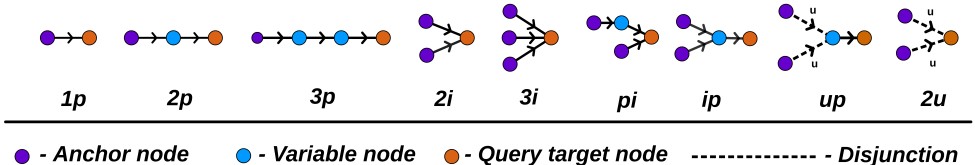

Figure 2: Query structures considered in our experiments, as proposed by Ren et al. (2020) – the naming of each query structure corresponds to *projection* (**p**), *intersection* (**i**), and *union* (**u**), and reflects how they were implemented in the Query2Box model (Ren et al., 2020). An example of a **pi** query is $?T : \exists V.p(a, V), q(V, T), r(b, T)$, where $a$ and $b$ are anchor nodes, $V$ is a variable node, and $T$ is the query target node.

Table 1: Number of queries in the datasets used for evaluation of query answering performance. Others indicates the number of queries for each of the remaining types.

| Dataset | Training | | Validation | | Test | |
|---|---|---|---|---|---|---|
| | 1p | Others | 1p | Others | 1p | Others |
| FB15k | 273,710 | 273,710 | 59,097 | 8,000 | 67,016 | 8,000 |
| FB15k-237 | 149,689 | 149,689 | 20,101 | 5,000 | 22,812 | 5,000 |
| NELL995 | 107,982 | 107,982 | 16,927 | 4,000 | 17,034 | 4,000 |

## 5 EXPERIMENTS

We described a method to answer a query by decomposing it into a continuous formulation, which we refer to as Continuous Query Decomposition (CQD). In this section we demonstrate the effectiveness of CQD on the task of answering complex queries that cannot be answered using the incomplete KG, and report experimental results for continuous optimisation (CQD-CO, Section 3.1) and beam search (CQD-Beam, Section 3.2). We also provide a qualitative analysis of how our method can be used to obtain explanations for a given complex query answer. For the sake of comparison, we use the same datasets and evaluation metrics as Ren et al. (2020).

### 5.1 DATASETS

Following Ren et al. (2020), we evaluate our approach on FB15k (Bordes et al., 2013) and FB15k-237 (Toutanova & Chen, 2015) – two subset of the Freebase knowledge graph – and NELL995 (Xiong et al., 2017), a KG generated by the NELL system (Mitchell et al., 2015). In order to compare with previous work on query answering, we use the queries generated by Ren et al. (2020) from these datasets. Dataset statistics are detailed in Table 1. We consider a total of 9 query types, including atomic queries, and 2 query types that contain disjunctions – the different query types are shown in Fig. 2. Note that in our framework, the neural link predictor is only trained on atomic queries, while the evaluation is carried out on the complete set of query types in Fig. 2.

Note that each query in Table 1 can have multiple answers, therefore the total number of training instances can be higher. For atomic queries (of type 1p), this number is equal to the number of edges in the training graph. Other methods like GQE (Hamilton et al., 2018) and Q2B (Ren et al., 2020) require a dataset with more query types. As an example, the FB15k dataset contains approximately 960k instances for 1p queries. When adding 2p, 3p, 2i, and 3i queries employed by GQE and Q2B during training, this number increases to 65 million instances.

### 5.2 MODEL DETAILS

To obtain embeddings for the query answering task, we use ComplEx (Trouillon et al., 2016) a variational approximation of the nuclear tensor $p$-norm for regularisation (Lacroix et al., 2018). We fix a learning rate of 0.1 and use the Adagrad optimiser. We then tune the hyperparameters of ComplEx on the validation set for each dataset, via grid search. We consider ranks (size of the

Table 2: Complex query answering results (H@3) across all query types; results for Graph Query Embedding (GQE, Hamilton et al., 2018) and Query2Box (Ren et al., 2020) are from Ren et al. (2020).

| Method | Avg | 1p | 2p | 3p | 2i | 3i | ip | pi | 2u | up |
|--------|-----|----|----|----|----|----|----|----|----|----|
| **FB15k** | | | | | | | | | | |
| GQE | 0.384 | 0.630 | 0.346 | 0.250 | 0.515 | 0.611 | 0.153 | 0.320 | 0.362 | 0.271 |
| Query2Box | 0.484 | 0.786 | 0.413 | 0.303 | 0.593 | 0.712 | 0.211 | 0.397 | 0.608 | 0.330 |
| CQD-CO | 0.576 | **0.918** | 0.454 | 0.191 | **0.796** | **0.837** | 0.336 | 0.513 | 0.816 | 0.319 |
| CQD-Beam | **0.680** | **0.918** | **0.779** | **0.577** | **0.796** | **0.837** | **0.375** | **0.658** | **0.839** | **0.345** |
| **FB15k-237** | | | | | | | | | | |
| GQE | 0.230 | 0.405 | 0.213 | 0.153 | 0.298 | 0.411 | 0.085 | 0.182 | 0.167 | 0.160 |
| Query2Box | 0.268 | 0.467 | 0.240 | 0.186 | 0.324 | 0.453 | 0.108 | 0.205 | 0.239 | **0.193** |
| CQD-CO | 0.272 | **0.512** | 0.213 | 0.131 | **0.352** | **0.457** | **0.146** | 0.222 | 0.281 | 0.132 |
| CQD-Beam | **0.290** | **0.512** | **0.288** | **0.221** | **0.352** | **0.457** | 0.129 | **0.249** | **0.284** | 0.121 |
| **NELL995** | | | | | | | | | | |
| GQE | 0.248 | 0.417 | 0.231 | 0.203 | 0.318 | 0.454 | 0.081 | 0.188 | 0.200 | 0.139 |
| Query2Box | 0.306 | 0.555 | 0.266 | 0.233 | 0.343 | 0.480 | 0.132 | 0.212 | 0.369 | 0.163 |
| CQD-CO | 0.368 | **0.667** | 0.265 | 0.220 | **0.410** | **0.529** | **0.196** | **0.302** | **0.531** | **0.194** |
| CQD-Beam | **0.375** | **0.667** | **0.350** | **0.288** | **0.410** | **0.529** | 0.171 | 0.277 | **0.531** | 0.156 |

embedding) in $\{100, 200, 500, 1000\}$, batch size in $\{100, 500, 1000\}$, and regularisation coefficients in the interval $\left[10^{-4}, 0.5\right]$.

For query answering we experimented with the Gödel and product t-norms – we select the best t-norm for each query type according to the best validation accuracy. For CQD-CO, we optimise variable and target embeddings with Adam, using the same initialisation scheme as Lacroix et al. (2018), with an initial learning rate of 0.1 and a maximum of 1,000 iterations. In practice, we observed that the procedure usually converges in less than 300 iterations. For CQD-Beam, the beam size $k \in \{2^2, 2^3, \ldots, 2^8\}$ is found on an held-out validation set.

## 5.3 EVALUATION

As in Ren et al. (2020), for each test query, we assign a score to every entity in the graph, and use such score for ranking such entities. We then compute the *Hits at 3* (H@3) metric, which measures the frequency with which the correct answer is contained in the top three answers in the ranking. Since a query can have multiple answers, we use the *filtered* setting (Bordes et al., 2013), where we filter out other correct answers from the ranking before calculating the H@3.

As baselines we use two recent state-of-the-art models for complex query answering, namely Graph Query Embedding (GQE, Hamilton et al., 2018) and Query2Box (Q2B, Ren et al., 2020).

## 5.4 RESULTS

We detail the results of H@3 for all different query types in Table 2. We observe that, on average, CQD produces more accurate results than GQE and Q2B, while using orders of magnitude less training data. In particular, combinatorial optimisation in CQD-Beam consistently outperforms the baselines across all datasets.

The results for chained queries (*2p* and *3p*) show that CQD-Beam is effective, even when increasing the length of the chain. The most difficult case corresponds to 3p queries, where the number of candidate variable substitutions increases due to the branching factor of the search procedure.

We also note that having more variables does not always translate into worse performance for CQD-CO: it yields the best ranking scores for *ip* queries on FB15k-237, and for *ip* and *pi* queries for NELL995, and both such query types contain two variables.

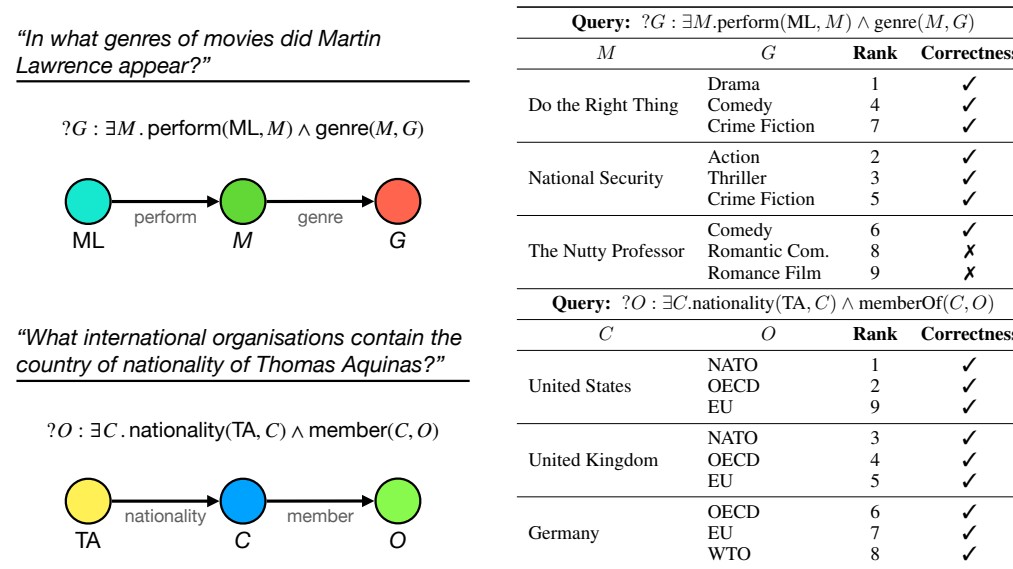

*"In what genres of movies did Martin Lawrence appear?"*

$?G : \exists M.\,\text{perform}(\text{ML}, M) \land \text{genre}(M, G)$

| **Query:** $?G : \exists M.\text{perform}(\text{ML}, M) \land \text{genre}(M, G)$ | | | |
|---|---|---|---|
| $M$ | $G$ | **Rank** | **Correctness** |
| Do the Right Thing | Drama | 1 | ✓ |
| | Comedy | 4 | ✓ |
| | Crime Fiction | 7 | ✓ |
| National Security | Action | 2 | ✓ |
| | Thriller | 3 | ✓ |
| | Crime Fiction | 5 | ✓ |
| The Nutty Professor | Comedy | 6 | ✓ |
| | Romantic Com. | 8 | ✗ |
| | Romance Film | 9 | ✗ |

*"What international organisations contain the country of nationality of Thomas Aquinas?"*

$?O : \exists C.\,\text{nationality}(\text{TA}, C) \land \text{member}(C, O)$

| **Query:** $?O : \exists C.\text{nationality}(\text{TA}, C) \land \text{memberOf}(C, O)$ | | | |
|---|---|---|---|
| $C$ | $O$ | **Rank** | **Correctness** |
| United States | NATO | 1 | ✓ |
| | OECD | 2 | ✓ |
| | EU | 9 | ✓ |
| United Kingdom | NATO | 3 | ✓ |
| | OECD | 4 | ✓ |
| | EU | 5 | ✓ |
| Germany | OECD | 6 | ✓ |
| | EU | 7 | ✓ |
| | WTO | 8 | ✓ |

Figure 3: Intermediate variable assignments and ranks for two example queries, obtained with CQD-Beam. Correctness indicates whether the answer belongs to the ground-truth set of answers.

The results presented in Table 2 were obtained with a rank of 1,000, as they produced the best performance in the validation set. We present results for other values of the rank in Appendix A, where we observe that even with a rank of 100, CQD still outperforms baselines with a larger embedding size. Furthermore, in Appendix B, we report the number of seconds required to answer each query type, showing that CQD-Beam requires less than 50ms for all considered queries.

We also experimented with a variant of CQD-Beam that uses DistMult (Yang et al., 2015) as the link predictor – results are reported in Appendix C. As expected, results when using DistMult are slightly less accurate than when using ComplEx, while still being more accurate than those produced by GQE and Q2B.

## 5.5 EXPLAINING ANSWERS TO COMPLEX QUERIES

A useful property of our framework is its transparency when computing scores for distinct atoms in a query. Unlike GQE and Q2B – two neural models that encode a query into a vector via a set of non-linear transformations – our framework is able to produce an explanation for a given answer in terms of intermediate variable assignments.

Consider the following test query from the FB15k-237 knowledge graph: "*In what genres of movies did Martin Lawrence appear?*" This query can be formalised as $?G : \exists M.\text{perform}(\text{ML}, M) \land \text{genre}(M, G)$, where ML is an anchor node representing Martin Lawrence. The ground truth answers to this query are 7 genres, including Drama, Comedy, and Crime Fiction. In Fig. 3 we show the intermediate assignments obtained when using CQD-Beam, to the variable $M$ in the query, and the rank for each combination of movie $M$ and genre $G$. We note that the assignments to the variable $M$ are correct, as these are movies where Martin Lawrence appeared. Furthermore, these intermediate assignments lead to correct answers in the first seven positions of the ranking, which correctly belong to the ground-truth set of answers.

In a second example, consider the following query: "*What international organisations contain the country of nationality of Thomas Aquinas?*" Its conjunctive form is $?O : \exists C.\text{nationality}(\text{TA}, C) \land \text{memberOf}(C, O)$, where TA is an anchor node representing Thomas Aquinas. The ground-truth answers to this query are the Organisation for Economic Co-operation and Development (OECD), the European Union (EU), the North Atlantic Treaty Organisation (NATO), and the World Trade Organisation (WTO). As shown in Fig. 3, CQD-Beam yields the correct answers in the first four positions in the rank. However, by inspecting the intermediate assignments, we note that such correct answers are produced by an incorrect (although related) intermediate assignment, since the country

of nationality of Thomas Aquinas is Italy. By inspecting these decisions we can thus identify failure modes of our framework, even when it produces seemingly correct answers. This is in contrast with other neural black-box models for complex query answering outlined in Section 4, where such an analysis is not possible.

## 6 CONCLUSIONS

We proposed a framework — Complex Query Decomposition (CQD) — for answering Existential Positive First-Order logical queries by reasoning over sets of entities in embedding space. In our framework, answering a complex query is reduced to answering each of its sub-queries, and aggregating the resulting scores via t-norms. The benefit of the method is that we only need to train a neural link prediction model on atomic queries to use our framework for answering a given complex query, without the need of training on millions of generated complex queries. This comes with the added value that we are able to explain each step of the query answering process regardless of query complexity, instead of using a black-box neural query embedding model.

The proposed method is agnostic to the type of query, and is able to generalise without explicitly training on a specific variety of queries. Experimental results show that CQD produces significantly more accurate results than current state-of-the-art complex query answering methods on incomplete Knowledge Graphs.

### ACKNOWLEDGEMENTS

This research was supported by the European Union's Horizon 2020 research and innovation programme under grant agreement no. 875160. This project was partially funded by Elsevier's Discovery Lab. Finally, we thank NVIDIA for GPU donations.

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

# A Influence of the Embedding Size on the Results

Table 3: Complex query answering results (H@3) across all query types, for different rank (embedding size) values – results for Graph Query Embedding (GQE, Hamilton et al., 2018) and Query2Box (Ren et al., 2020) are from Ren et al. (2020).

| Method | Rank | 1p | 2p | 3p | 2i | 3i | ip | pi | 2u | up |
|---|---|---|---|---|---|---|---|---|---|---|
| **FB15k** | | | | | | | | | | |
| GQE | 800 | 0.630 | 0.346 | 0.250 | 0.515 | 0.611 | 0.153 | 0.320 | 0.362 | 0.271 |
| Query2Box | 400 | 0.786 | 0.413 | 0.303 | 0.593 | 0.712 | 0.211 | 0.397 | 0.608 | 0.330 |
| CQD-CO | 100 | 0.893 | 0.162 | 0.076 | 0.773 | 0.818 | 0.118 | 0.344 | 0.493 | 0.073 |
| | 200 | 0.906 | 0.257 | 0.092 | 0.785 | 0.828 | 0.210 | 0.426 | 0.753 | 0.110 |
| | 500 | 0.912 | 0.345 | 0.123 | 0.772 | 0.817 | 0.257 | 0.454 | 0.795 | 0.206 |
| | 1000 | **0.918** | 0.454 | 0.191 | **0.796** | **0.837** | 0.336 | 0.513 | 0.816 | 0.319 |
| CQD-Beam | 100 | 0.893 | 0.746 | 0.557 | 0.773 | 0.818 | 0.357 | 0.669 | 0.689 | 0.313 |
| | 200 | 0.906 | 0.770 | **0.585** | 0.785 | 0.828 | 0.373 | 0.679 | 0.815 | **0.357** |
| | 500 | 0.912 | 0.759 | 0.580 | 0.772 | 0.817 | 0.372 | 0.650 | 0.831 | 0.351 |
| | 1000 | **0.918** | **0.779** | 0.584 | **0.796** | **0.837** | **0.377** | **0.658** | **0.839** | 0.355 |
| **FB15k-237** | | | | | | | | | | |
| GQE | 800 | 0.405 | 0.213 | 0.153 | 0.298 | 0.411 | 0.085 | 0.182 | 0.167 | 0.160 |
| Query2Box | 400 | 0.467 | 0.240 | 0.186 | 0.324 | 0.453 | 0.108 | 0.205 | 0.239 | **0.193** |
| CQD-CO | 100 | 0.493 | 0.162 | 0.076 | 0.311 | 0.415 | 0.118 | 0.199 | 0.238 | 0.073 |
| | 200 | 0.500 | 0.187 | 0.092 | 0.329 | 0.439 | 0.128 | 0.204 | 0.254 | 0.103 |
| | 500 | 0.508 | 0.210 | 0.123 | 0.346 | 0.454 | 0.142 | 0.216 | 0.273 | 0.119 |
| | 1000 | **0.512** | 0.213 | 0.131 | **0.352** | **0.457** | **0.146** | 0.222 | 0.281 | 0.132 |
| CQD-Beam | 100 | 0.493 | 0.256 | 0.207 | 0.311 | 0.415 | 0.119 | 0.234 | 0.254 | 0.121 |
| | 200 | 0.500 | 0.272 | 0.216 | 0.329 | 0.439 | 0.122 | 0.244 | 0.264 | 0.127 |
| | 500 | 0.508 | **0.280** | 0.216 | 0.346 | 0.454 | 0.127 | **0.257** | 0.280 | 0.128 |
| | 1000 | **0.512** | 0.279 | **0.219** | **0.352** | **0.457** | 0.129 | 0.249 | **0.284** | 0.128 |
| **NELL995** | | | | | | | | | | |
| GQE | 800 | 0.417 | 0.231 | 0.203 | 0.318 | 0.454 | 0.081 | 0.188 | 0.200 | 0.139 |
| Query2Box | 400 | 0.555 | 0.266 | 0.233 | 0.343 | 0.480 | 0.132 | 0.212 | 0.369 | 0.163 |
| CQD-CO | 100 | 0.647 | 0.234 | 0.145 | 0.389 | 0.508 | 0.165 | 0.283 | 0.465 | 0.126 |
| | 200 | 0.658 | 0.238 | 0.164 | 0.401 | 0.524 | 0.172 | 0.282 | 0.502 | 0.148 |
| | 500 | 0.665 | 0.261 | 0.208 | 0.406 | 0.525 | 0.187 | 0.293 | 0.523 | 0.171 |
| | 1000 | **0.667** | 0.265 | 0.220 | **0.410** | **0.529** | 0.196 | 0.302 | 0.531 | **0.194** |
| CQD-Beam | 100 | 0.647 | 0.333 | 0.296 | 0.389 | 0.508 | 0.160 | 0.293 | 0.469 | 0.150 |
| | 200 | 0.658 | 0.335 | 0.292 | 0.401 | 0.524 | 0.162 | 0.290 | 0.508 | 0.146 |
| | 500 | 0.665 | **0.348** | 0.296 | 0.406 | 0.525 | 0.166 | 0.291 | 0.527 | 0.149 |
| | 1000 | **0.667** | 0.343 | **0.297** | **0.410** | **0.529** | 0.168 | 0.283 | **0.536** | 0.157 |

In Table 3 we report results for CQD-CO (Section 3.1) and CQD-Beam (Section 3.2) for different rank (embedding size) values. We can see that the model produces very accurate results even with significantly fewer parameters.

## B    TIMING EXPERIMENTS

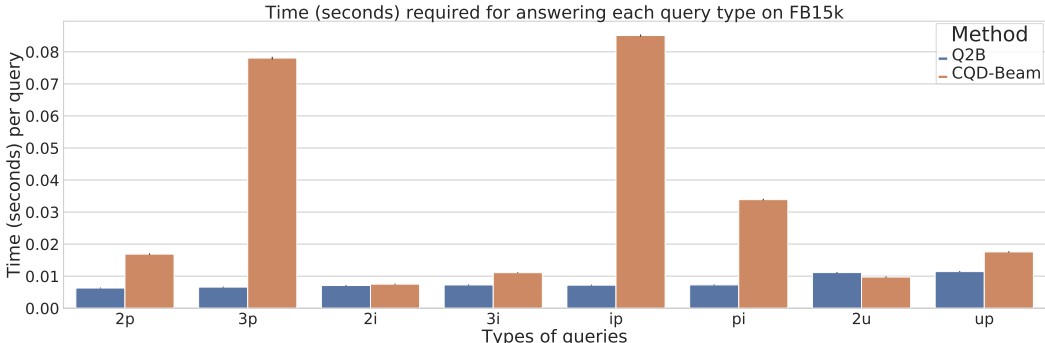

Figure 4: Number of seconds required by Q2B (Ren et al., 2020) and CQD-Beam (Section 3.2 for answering each query type in FB15k.

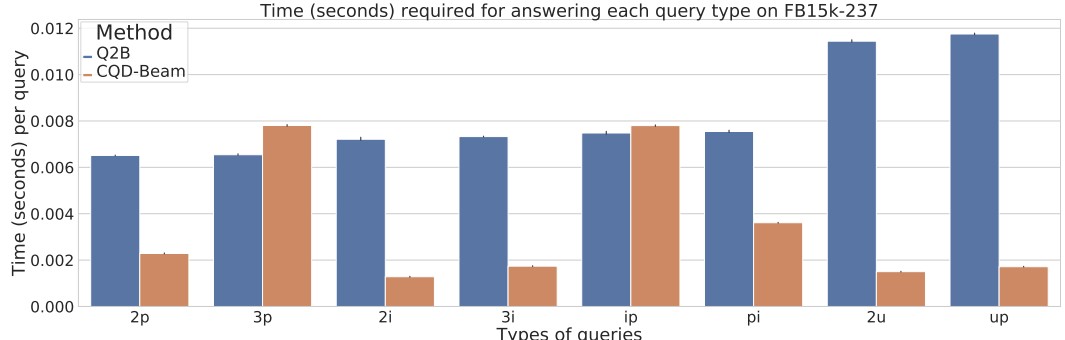

Figure 5: Number of seconds required by Q2B (Ren et al., 2020) and CQD-Beam (Section 3.2 for answering each query type in FB15k-237.

In Fig. 4 and Fig. 5 we report the time (seconds) required by Q2B (Ren et al., 2020) and CQD-Beam (Section 3.2 for answering each query type, aggregated over FB15k, FB15k-237, and NELL. We can see that, in CQD-Beam, the main computation bottleneck are multi-hop queries, since the model is required to invoke the neural link prediction model for each step of the chain to obtain the top-$k$ candidates for the next step in the chain.

## C  DISTMULT EXPERIMENTS

Table 4: Complex query answering results (H@3) across all query types, for two different neural link prediction models, namely ComplEx (Trouillon et al., 2016) and DistMult (Yang et al., 2015).

| Method | Model | 1p | 2p | 3p | 2i | 3i | ip | pi | 2u | up |
|--------|-------|------|------|------|------|------|------|------|------|------|
| **FB15k** | | | | | | | | | | |
| CQD-Beam | ComplEx | 0.918 | 0.779 | 0.584 | 0.796 | 0.837 | 0.377 | 0.658 | 0.839 | 0.355 |
| | DistMult | 0.869 | 0.761 | 0.581 | 0.778 | 0.824 | 0.369 | 0.608 | 0.822 | 0.355 |
| **FB15k-237** | | | | | | | | | | |
| CQD-Beam | ComplEx | 0.512 | 0.279 | 0.219 | 0.352 | 0.457 | 0.129 | 0.249 | 0.284 | 0.128 |
| | DistMult | 0.485 | 0.277 | 0.210 | 0.332 | 0.443 | 0.117 | 0.224 | 0.281 | 0.123 |
| **NELL995** | | | | | | | | | | |
| CQD-Beam | ComplEx | 0.667 | 0.343 | 0.297 | 0.410 | 0.529 | 0.168 | 0.283 | 0.536 | 0.157 |
| | DistMult | 0.642 | 0.348 | 0.297 | 0.392 | 0.517 | 0.160 | 0.260 | 0.502 | 0.169 |

In Table 4 we report the results for CQD-Beam with two different neural link prediction models, namely ComplEx (Trouillon et al., 2016) and DistMult (Yang et al., 2015). Both models were trained using the loss and regulariser proposed by Lacroix et al. (2018), and their hyperparameters were tuned according to their performance in the validation set; in both cases, the embedding size is set to 1,000. As expected, CQD-Beam with DistMult produces slightly less accurate results than with ComplEx, while still yielding more accurate results than the Q2B and GQE baselines.

