# OpenReview forum: "Complex Query Answering with Neural Link Predictors"
_ICLR.cc/2021/Conference — ICLR 2021 Oral_

### Official Review · AnonReviewer4 · 2020-10-24
**nice improvement of the SOTA**

**Rating:** 9
**Confidence:** 4

**Review:**

The paper proposes Continuous Query Decomposition (CQD), an approach for answering Existential Positive First-Order (EPFO)
queries over incomplete knowledge graphs exploiting a neural link predictor for 1-hop-only queries.
Entities are embedded in a low dimensional space and entity vectors are used to compute the score of query atoms that
are then combined using a t-norm for conjunction and t-conorm for disjunction.
Answers to queries are found either with continuous optimisation by gradient descent to find embeddings for query variables
or combinatorial optimisation where top-k entities for query variables are looked for yielding a beam search.
CQD is compared with Graph Query Embedding (GQE) and Query2Box over three datasets on a large number of queries.
The result show that CQD outperforms the baselines on Hit@3 on average.
CQD also offers the possibility of explaining the results of queries by showing the top scoring entities for query variables and the score of atoms.

CQD tackles the difficult problem of answering queries that are beyond simple 1-hop completion queries. It improves
over previous work which need to train the model over a large number of queries (Hamilton et al., 2018;Daza & Cochez, 2020;
Ren et al., 2020) and do not consider disjunctive queries (Hamilton et al., 2018; Daza & Cochez, 2020).
These advantages are obtained by not embedding the query into a low dimensional space but using continuous or combinatorial
optimization to answer queries, considering the query as a formula in fuzzy logic and applying t-norms and t-conorms.
While the use of fuzzy logic in query answering is not new, they way in which it is combined with entity embeddings and
neural link predictors is original to the best of my knowledge.

The fact that queries are not embedded (and so learning does not need large numbers of queries) is a strong point of CQD,
with competing methods (Hamilton et al., 2018;Daza & Cochez, 2020; Ren et al., 2020) requiring many queries for tuning the query embeddings.
Since queries are not embedded, the results of CQD are also easier to explain.

The experiments are sufficiently extensive to support the claim of the paper that CQD is also outperforming competitors in
terms of the quality of solutions. However, the authors should justify why they used embedding size 500 for their methods
and 400 for the baselines.

From a technical point of view the article seems sound but the authors say that "Then, after we identified the optimal
representation for variables $A, V_1, \ldots  V_m$, we replace the query target embedding $e_A$ with the embedding
representations $e_c \in R^k$ of all entities $c \in E$, and use the resulting complex query score to compute the
likelihood that such entities answer the query."
In this way the authors throw away vector $e_A$ that may have information about the problems, isn't there a method to
exploit the information in $e_A$?

I have a few remarks about the presentation:
Citation Raedt, 2008 should be De Raedt, 2008.
In Figure 1 the edges of the graphs have the opposite direction with respect to the caption and main text.
Page 6: "we only make use of type 1-chain queries to train the neural link
predictor": do the authors mean 1-hop queries? 1-chain appears here for the first time.
"In all cases, we use a rank of 500.": for rank do the authors mean the embedding size? This should be clarified
Page 7: "Since
a query can have multiple answers, we implement a filtered setting, whereby for a given answer, we
filter out other correct answers from the ranking before computing H@3.": this sentence is not clear. Does it mean
that answers that follow from the KG without completion are removed from the ranking?

----After reading the other reviews and the authors' comments, I still think the paper is excellent and should be accepted.

---

> ### Author Response · Authors · 2020-11-12
> **Response to Reviewer 4**
>
> Thank you for your valuable comments and feedback. Please find our response to your questions next. We did incorporate your remarks about the presentation in the updated version.
>
> - 1-chain appears here for the first time. "In all cases, we use a rank of 500.": for rank do the authors mean the embedding size?
>
> Yes! We do refer to the embedding size as the rank - we clarified this in the updated version.
>
> We searched for the optimal embedding size by tuning it on a held-out validation set - we now provide a detailed description of the hyperparameter search process, as well as results with different embedding sizes (see Appendix, Sec. A).
> Overall, we note that even with a lower rank of 100, our method still produces more accurate ranking results than baselines with larger embedding sizes (GQE and Q2B).
>
> - From a technical point of view the article seems sound but the authors say that "Then, after we identified the optimal representation for variables A,V_1,…V_m, we replace the query target embedding e_A with the embedding representations e_c∈Rk of all entities c∈E, and use the resulting complex query score to compute the likelihood that such entities answer the query." [..] isn't there a method to exploit the information in eA?
>
> Indeed, we discard the embedding of the target variable, because ultimately our aim is to score actual entities in the KG.
> In a previous iteration of the project, we were ranking entities according to their distance from the vector e_A. However, we quickly realised this makes strong assumptions on the geometry of the embedding space induced by the neural link predictor, and also produced less accurate results.
>
> - Page 7: "Since a query can have multiple answers, we implement a filtered setting, whereby for a given answer, we filter out other correct answers from the ranking before computing H@3.": this sentence is not clear. Does it mean that answers that follow from the KG without completion are removed from the ranking?
>
> We use the same evaluation protocol as GQE and Q2B: when ranking the candidate answers for a query and the gold answer is x, we remove the other entities that correctly answer the query and are different from x. This setting is used for not penalising the model for ranking other correct answers higher than x, since all these answers are valid.

---

> > ### Comment · AnonReviewer4 · 2020-11-16
> > **thanks for the clarifications**
> >
> > Your comment cleared my doubts

---

### Official Review · AnonReviewer1 · 2020-10-27
**Surprisingly simple idea that seems to work**

**Rating:** 8
**Confidence:** 4

**Review:**

The paper attempt to answer conjunctive queries that are in the form of a chain of facts bound together with unobserved variables. The authors suggest that you can use any relational learning method to embed entities and relations in a k-dimensional space and then use the t-norm in order to create a loss function that will be used in order to find the result of the query.  The paper investigates continuous optimization through stochastic gradient descent and a greedy method for combinatorial optimization. The results demonstrate that the greedy optimization method performs better. In addition, they claim that their method outperforms other methods with the advantage of using less training data.

Here are some comments
* I think the authors cover the relevant work sufficiently
* The idea is very simple and builds upon other work that is well studied and well understood by the community
* I think that there is an excessive mathematical formalism that is a bit unnecessary. There is no reason for that, the fact that the idea is very simple does not mean that we have to add extra formalism.
* In terms of generalization, I think it is very interesting that the users train only on 1-hop  queries and evaluate up to 5-hop. In the introduction, the authors claim they use less data than the other methods, but they don’t make it very clear in the experimental section. I think they need to be more explicit about that. They need to clarify that less data means just the 1-hop queries
* I have two reservations about the paper. I suspect that part of the success of their approach is the ComplEx embeddings. I would appreciate an ablation study with at least one more method for relational learning, let’s say TransE to see how sensitive it is on the embeddings. To be fair the authors study in depth the performance of their algorithm in other variations, such as the length of the chain
* The other concern is about timing results. It would help to know how the whole algorithm compares in terms of time to answer the query compared to the others. I think it is of particular interest the difference between the two optimization techniques. I suspect that the greedy one might be two slow for longer chains. From the preliminary analysis, it seems to grow as k^d where k is the width of the beam per relation and d is the length of the chain.

In general, I think it is a very practical paper

---

> ### Author Response · Authors · 2020-11-12
> **Response to Reviewer 1**
>
> Thank you for your questions and valuable feedback.
>
> - I think that there is an excessive mathematical formalism that is a bit unnecessary. There is no reason for that, the fact that the idea is very simple does not mean that we have to add extra formalism.
>
> Indeed, we agree with you: we tried our best to explain our method as simple as possible (by providing plenty of examples and visual intuitions), while still using the same terminology as related work in this area (to unambiguously specify what kind of queries our method can answer), and without loss in generality.
> We also think that the notation allows us to conveniently re-state the problem as an optimisation problem where scores are computed using t-norms and t-conorms.
> Which parts do you think that can be improved in terms of clarity?
>
> - In the introduction, the authors claim they use less data than the other methods, but they don’t make it very clear in the experimental section. I think they need to be more explicit about that. They need to clarify that less data means just the 1-hop queries.
>
> Thank you for pointing this out, we agree that we could have been more explicit about the fact that our method only requires 1-hop queries for training: we have added details about this and actual numbers to contrast with the amount of data required by other methods.
>
> - I have two reservations about the paper. I suspect that part of the success of their approach is the ComplEx embeddings.
>
> We agree that ComplEx is a significant contributor to the statistical accuracy in our model, which we chose since it is a fairly simple but still extremely competitive neural link predictor [1]. As an additional analysis, in the updated version of the paper (Tab. 3) we report results with ComplEx with different rank values (embedding sizes), showing we can significantly reduce the embedding size in the underlying neural link predictor without losing too much in ranking accuracy. Furthermore, we are now in the process of running additional experiments with DistMult, another neural link prediction model, which should be ready in the next two days.
>
> [1] “You CAN Teach an Old Dog New Tricks! On Training Knowledge Graph Embeddings“ - ICLR 2019, https://openreview.net/forum?id=BkxSmlBFvr
>
> - The other concern is about timing results. It would help to know how the whole algorithm compares in terms of time to answer the query compared to the others. I think it is of particular interest the difference between the two optimization techniques. I suspect that the greedy one might be two slow for longer chains. From the preliminary analysis, it seems to grow as $k^d$ where k is the width of the beam per relation and d is the length of the chain.
>
> Thank you for pointing this out as well - we just included accurate timing results in the updated version of the paper (Appendix, Sect. B). We found that the time required for the combinatorial optimisation is on par or higher than Q2B, but always below 50ms per query.
> Indeed the greedy algorithm tends to be slower on longer chains: the neural link predictor is invoked once for each of the hops in the chain, for obtaining a list of top-k candidates to use in the next step in the chain. In our experiments, identifying the top-k candidates for each step in a chain was not an issue, since all candidate entities can be scored in parallel very efficiently on GPU.
> A potential bottleneck in terms of space complexity can be the number of candidate variable assignments, which at the moment is given by $k^d$ (we did not experience any issues related to this, since in the datasets we considered $d$ is at most 3). A solution for handling longer chains may consist in trading complexity with completeness, and e.g. set an upper bound to the number of candidate variable assignments being considered.

---

### Official Review · AnonReviewer2 · 2020-10-29
**A new method on reasoning on KG, nice empirical results**

**Rating:** 6
**Confidence:** 5

**Review:**

The paper aims to answer complex queries on knowledge graphs. Different from previous methods that aim to embed the queries, the method views the query answering problem as an optimization / search problem where the goal is to find the most plausible entities on the reasoning path. The merits are that the method only needs to train on 1 hop path queries (link prediction), saving the effort of training on complex queries as in previous work, and proposes two solutions, which both achieve nice results on standard multi-hop reasoning benchmarks. It also demonstrates interpretability of the model by showing some examples of the intermediate entities found in the reasoning path when answering a complex multi-hop query.

I think the paper is clear and easy to follow. I have some questions regarding the two methods. For the first method, continuous optimization in sec 3.1, what is the difference between this method and the previous works GQE, Q2B, etc. apart from different neural link predictors? Especially for path queries, e.g., given a two hop query, $(\text{Obama}, \text{BornIn}, V_1)\wedge(V_1, \text{CapitalOf}, V_2)$, then the optimal $e_{V_1}$ will be $e_{Obama}+e_{BornIn}$, because the distance will be 0, and $\phi_p$ will be 1 (here it assumes TransE model, and of course it can be generalized to DisMult, ComplEx, etc.). Then the first formulation is in essence very much similar to GQE, because GQE/Q2B also models $\mathbf{e}_{V_1}$ in the exact same way and the difference only lies in (1) you use ComplEx (2) t-norm modeling of conjunction? However, it seems that t-norm demonstrates less expressiveness for modeling conjunction because both GQE/Q2B models conjunction using a MLP with additional learnable parameters, which can also approximate t-norm and even be more adaptive depending on the training queries/KG.
For the second method, the time complexity seems exponential with respect to the number of hops. For a m hop query, and each step you keep the top-k, then do you end up with $k^m$ entities?

Additional questions:
1. How did you calibrate the output of ComplEx so that $\phi_p(e_s, e_o)$ is in [0,1]? Better to add more details on neural link predictors.
2. Some ablation studies that use different t-norm and t-conorm other than the Godel and product may make the argument stronger.
3. There exists a tradeoff between the inference time and training queries. For GQE/Q2B, they can leverage complex queries to train the conjunction operator (MLP), so that during inference, there is no need to do any optimization. But for the proposed method, it saves the effort of training on complex queries, however, during inference, the method needs an online optimization process to instantiate the variables on the path. Especially for CQD-CO, the authors mention that they need to optimize online for 1000 iterations, which is too expensive for answering a query. Can you list the inference time of both models (continuous, combinatorial) and compare it with GQE/Query2box?
4. For 1p performance, it is equivalent to the performance of ComplEx on link prediction right?
5. The table 3 is confusing, why are the numbers (e.g., 5.5, 46.76) larger than 1, I think the model normalized the output of $\phi$ to [0,1]?
6. The proposed two optimization methods are independent of neural link predictors. Can you use the same neural link predictor for your models and GQE for fair comparison? You can train a TransE model for the neural link predictor, and accordingly define $\phi_p$, then it will be clear to show whether the gain comes from a different neural link predictor (TransE vs ComplEx), or comes from the t-norm and the two optimization methods. And of course another choice is the other way around, e.g., use the ComplEx version of GQE and make the same comparison.

Minor points to fix:
1. In the method section, bold $\mathbf{e}$ denotes vector embedding while normal $e$ denotes a logic formula, which is subtle and confusing. Authors can change the notation of one of them.
2. Also, the notation $e_i^j$ is abused, in Eq. 2, it represents a logic formula, however, in Eq. 3, it represents the output of $\phi_p$, which is a scalar.

---

> ### Author Response · Authors · 2020-11-12
> **Response to Reviewer 2**
>
> Thank you for your questions and valuable feedback.
>
> - For the first method, continuous optimization in sec 3.1, what is the difference between this method and the previous works GQE, Q2B, etc. apart from different neural link predictors?
>
> Given a complex query, GQE and Q2B produce an embedding representation of such a query and use it for ranking all candidate answers according to a matching score between the query and the answer embeddings. On the other hand, our model decomposes a complex query into simpler (atomic) queries, which are answered individually using a neural link predictor, and then intermediate scores are aggregated using t-norms and t-conorms -- continuous relaxations of the logical conjunction and disjunction operators.
>
> By doing so, we are also able to produce explanations for why a given answer was selected in terms of the intermediate answers for the atomic queries -- we elaborate on this aspect in the updated version of this paper.
>
> You mention that “Especially for path queries, e.g., given a two-hop query, (Obama,BornIn,V1) ∧ (V1,CapitalOf,V2), then the optimal e_V1 will be eObama+eBornIn”. We completely agree with this: if we select TransE as our underlying neural link predictor, indeed CQD-CO would be quite related to the model you just proposed.
>
> However, doing the same for other neural link predictors such as DistMult and ComplEx is not as simple, since the values of e_V1 and e_V2 identified by optimising the query score would not be meaningful (it’s possible to maximise the query score by just increasing the norm of e_V1 and e_V2). Our aim is proposing a solution that is not model-dependent (i.e. that can be used with any neural link predictor); with interesting explainability properties; and does not require training on complex queries (we can aggregate intermediate results with t-norms and t-conorms, without learning additional parameters) while still achieving SOTA results.
>
> -  For the second method, the time complexity seems exponential with respect to the number of hops.
>
> Indeed, for a multi-hop query, the second method produces $k^m$ variable assignments -- this was not an issue in our experiments since in the complex query answering datasets considered by GQE and Q2B, the m in multi-hop queries is at most three. We argue that, for higher values of m, we can control the space complexity of the method by using a more space-efficient variant of the method, where the number of variable assignments is bounded to some constant.
>
> Furthermore, answering each hop for and identifying the top-k candidates can be done in constant time on GPU, since all candidate entities can be scored in parallel very efficiently (in ComplEx, the scoring function is a trilinear dot product, thus scoring all entities can be reduced to a matrix-vector multiplication).
> We updated our paper including actual timing measurements in the appendix.
>
> - How did you calibrate the output of ComplEx so that $\phi_{p}(e_{s}, e_{o}) is in [0,1]?
>
> We use the sigmoid to map ComplEx scores to values in [0, 1].
>
> - Can you list the inference time of both models (continuous, combinatorial) and compare it with GQE/Query2box?
>
> We included explicit timing results for combinatorial optimisation in the updated version of the paper - see Sect. B in the Appendix. For the continuous optimisation version, we found that it can take between 1 to 10 seconds to answer all the queries in FB15k, FB15k-237, and NELL, thanks to the fact that those operations can be efficiently parallelised on GPU.
>
> - For 1p performance, it is equivalent to the performance of ComplEx on link prediction right?
>
> Yes, the performance of our method on 1p (atomic) queries is determined by the accuracy of the neural link predictor. However, upon further investigation, we noticed that Q2B uses a different evaluation procedure for atomic queries: we updated the results table, and we find that our method produces more accurate results on atomic queries as well.
>
> - The table 3 is confusing, why are the numbers (e.g., 5.5, 46.76) larger than 1, I think the model normalized the output of $\phi$ to [0,1]?
>
> Thank you for pointing this out -- we reported the logits rather than the normalised values. We solved the issue in the updated version of the paper, and added an additional example.
>
> - The proposed two optimization methods are independent of neural link predictors. Can you use the same neural link predictor for your models and GQE for fair comparison?
>
> We chose ComplEx because it is a very simple yet extremely effective neural link predictor, but indeed it would be interesting to test our method with different models. To account for this, in the updated version of the paper we included experiments with different ranks for ComplEx - showing we can decrease the rank from 1000 to 100 without significantly decreasing the predictive accuracy of the model - and are now in the process of running additional experiments with DistMult (which is also considered in GQE).

---

> > ### Author Response · Authors · 2020-11-24
> > **Response to Reviewer 2 (2)**
> >
> > - The proposed two optimization methods are independent of neural link predictors. Can you use the same neural link predictor for your models and GQE for fair comparison?
> >
> > We just updated our submission with additional experiments on adopting DistMult rather than ComplEx as the underlying neural link prediction model - results are available in the appendix (Sect. C). We find that results with DistMult are slightly less accurate than with Complex, as we expected, while still more accurate than the GQE and Q2B baselines.
> >
> > A bilinear interaction model similar to DistMult was also considered as a projection operator by GQE, with significantly less accurate results. We believe that our improvements in terms of accuracy can be attributed not just to the use of a competitive neural link predictor, but also to the compositional nature of our model, which allows it to generalise from atomic to complex queries thanks to the use of t-norms and t-conorms.

---

### Official Review · AnonReviewer3 · 2020-10-29
**Complex logical query evaluation with link predictors**

**Rating:** 9
**Confidence:** 2

**Review:**

Summary:
This paper proposes Continuous Query Decomposition (CQD) a novel method for evaluating complex queries over incomplete KGs. Each variable of a logical query (involving existential quantifiers, conjunctions and disjunctions) is mapped to an embedding. A link predictor, trained on single edge prediction, is used to score the atomic query involving the variable. The full query is evaluated using continuous versions of the logical operators and gradient-based or combinatorial optimization.
Evaluating complex logical queries on (necessarily incomplete) KGs and other graph-structured data is an important problem for data mining purposes. The paper proposes an elegant and effective method.

Strong points
Elegant, efficient solution.
SOTA results.
Provides aspects of explainability, although this could be discussed and illustrated better.

Detailed comments
- What is particularly important/challenging about EPFO queries, beyond existential and conjunctive ones? Obviously it is an extension that covers more FOL, but a qualitative discussion would help the reader, particularly with respect to applications to KGs.
- Could you talk more, give more insights about the 8 complex queries types? Why are they important?
- The query “What international organisations contain the country of nationality of Thomas Aquinas?” sounds really artificial. Maybe there is a better example involving entities and relations, similar to the drugs one?
- Could you say a bit more with respect to how the KG incompleteness is accounted for in the evaluation?
- The paper mentions “.. in many complex domains, an open challenge is developing techniques for answering complex queries involving multiple and potentially unobserved edges, entities, and variables, rather than just single edges.” It would be great to articulate this more for sake of providing context and motivation.

---

> ### Author Response · Authors · 2020-11-12
> **Response to Reviewer 3**
>
> Thank you for your valuable feedback and comments! We next address your comments.
>
> - What is particularly important/challenging about EPFO queries, beyond existential and conjunctive ones?
>
> We consider a query as determined by a series of arbitrary constraints expressed in some language: a more expressive language makes it possible to answer a broader set of queries.
>
> The simplest case is link prediction, where the constraint is a single predicate, whereas recent methods have started considering existentially quantified variables and conjunctions of predicates [1] and disjunctions [2], which together with conjunctions form EPFO queries). Considering EPFO queries is thus a step towards answering increasingly more expressive queries.
>
> - Could you talk more, give more insights about the 8 complex queries types? Why are they important?
>
> The 8 query structures that we experiment with allow us to compare with other works in the complex query answering literature that use such queries for evaluation - see e.g. [1, 2].
>
> - The query “What international organisations contain the country of nationality of Thomas Aquinas?” sounds really artificial. Maybe there is a better example involving entities and relations, similar to the drugs one?
>
> That’s right - thank you for pointing this out: we added a more realistic example in the updated version of the paper.
>
> - Could you say a bit more with respect to how the KG incompleteness is accounted for in the evaluation?
>
> The queries we evaluated on are standard datasets proposed and used by e.g. [1, 2]: some edges are removed at random from the KG, and the queries are generated in such a way that one needs the missing edges in order to answer them. Then, a neural model is trained to answer the queries while accounting for the missing edges.
>
> [1] Hamilton et al. 2018, “Embedding Logical Queries on Knowledge Graphs”, NeurIPS 2018.
>
> [2] Ren et al. 2020, “Query2box: Reasoning over Knowledge Graphs in Vector Space using Box Embeddings”, ICLR 2020.

---

### Comment · ~Jiaxin_Bai1 · 2021-03-10
**Questions about the optimization methods**

I am a researcher that is interested in the area of complex query answering. However, the descriptions of the experiments are so vague that it is hard to reproduce the experiment according to the paper only. So I have to ask some questions here about the experimental settings and evaluation details.


1. Generally, the paper said only the atomic queries are used to train the model. What are atomic queries? Are they 1-projection queries? If the model is only trained on 1-projection queries, it is basically training a link predictor. Then why you describe two optimization methods for complex queries? This is really confusing. Or actually, the model was trained on multiple complex query types?

2. What is your inference method for your model? It seems that your model can be optimized in two ways as mentioned in the paper. But during the inference time, only the beam search method can be used to do inference. My question is whether both CQD-CO and CQD-Beam models are evaluated by the Beam search method.

3. Why not just use a pre-trained link predictor? It seems that a pre-trained link predictor can be also used to do inference by using beam search. Also, I think this might be the most appropriate baseline to evaluate the effectiveness of both optimization methods.

Although I am really confused by some descriptions in the paper, I really think this is a great paper. Because it points out an important new direction of solving complex query answering.


Will you release the code for the experiments?

---

> ### Comment · ~Pasquale_Minervini2 · 2021-03-16
> **Re: Questions about the optimization methods**
>
> Hi Jiaxin!
>
> > Generally, the paper said only the atomic queries are used to train the model. What are atomic queries? Are they 1-projection queries? If the model is only trained on 1-projection queries, it is basically training a link predictor. Then why you describe two optimization methods for complex queries? This is really confusing. Or actually, the model was trained on multiple complex query types?
>
> Yes, we train a neural link predictor to be able to answer atomic queries! The problem is that you still need to find the optimal variable assignments for each query -- depending on how you cast it, you can see either as a combinatorial optimisation problem (if you search for the optimal variable-to-entity mapping) or a continuous optimisation problem (in case you search for the optimal variable-to-entity embedding mapping), hence the two optimisation methods.
>
> > What is your inference method for your model?
>
> The two optimisation methods are used for inference!
>
> > Why not just use a pre-trained link predictor?
>
> We do that! :) We had to experiment with several configurations to find the optimal hyperparameters. Then we used the best models we trained for answering complex queries (we uploaded all models online, the link is available on the GitHub repo).
>
> > Will you release the code for the experiments?
>
> Yes, it's online! The link should be visible in the abstract.

---

### Comment · ~William_W._Cohen2 · 2021-04-02
**related work: Faithful Embeddings for Knowledge Base Queries**

I haven't read through your paper in detail yet - it looks very interesting! - but our NeurIPS paper https://arxiv.org/abs/2004.03658 is quite related - we also compared to Query2Box, and we also were able to get good performance by training only on simple queries.

---

> ### Comment · ~Pasquale_Minervini2 · 2021-04-02
> **Re: Faithful Embeddings for Knowledge Base Queries**
>
> Hi William! Somehow we managed to miss this one, it's probably the most related work I've seen so far -- thank you for sharing it, we will include it ASAP.

---

### Decision · Program_Chairs · 2021-01-07
**Final Decision**

**Decision:**

Accept (Oral)

**Comment:**

The reviewers unanimously agree that this paper is a strong accept; it makes important progress in developing our ability to query relational embedding  models.